# The N-Terminal Domain of Tailspike Depolymerases Affects the Replication Efficiency of Synthetic *Klebsiella* Phages

**DOI:** 10.3390/ijms262311297

**Published:** 2025-11-22

**Authors:** Ivan K. Baykov, Ekaterina E. Mikhaylova, Anna V. Miroshnikova, Valeriya A. Fedorets, Sofya A. Markova, Tatyana A. Ushakova, Vera V. Morozova, Nina V. Tikunova

**Affiliations:** 1Laboratory of Molecular Microbiology, Institute of Chemical Biology and Fundamental Medicine of the Siberian Branch of the Russian Academy of Sciences, Novosibirsk 630090, Russia; 2Shared Research Facility “Siberian Circular Photon Source” (SRF “SKIF”), Boreskov Institute of Catalysis of the Siberian Branch of the Russian Academy of Sciences, Novosibirsk 630090, Russia; 3Faculty of Natural Sciences, Novosibirsk State University, Novosibirsk 630090, Russia

**Keywords:** bacteriophage, *Klebsiella*, tailspike depolymerase, receptor-binding protein, anchor domain, adapter domain, synthetic biology, genome assembly, transformation-associated recombination cloning

## Abstract

Bacteriophage receptor-binding proteins are often attached to the tail via a conserved N-terminal adapter/anchor domain, presumed to function independently from the distal receptor-binding/catalytic domain. Using synthetic phage technology, we demonstrated that the N-terminal domain in *Przondovirus* phages KP192 and KP195 substantially modulates the receptor-binding and hydrolytic activities of their type A tailspikes. A bioinformatics analysis of related proteins revealed a high correlation between the N-terminal domain and the distal receptor-binding region. Furthermore, it was shown that an imperfect structural fit between the N-terminal domain and the adjacent tail proteins (gatekeeper and nozzle proteins) can reduce virion assembly efficiency, thereby impairing phage fitness. These results underscore the importance of selecting an appropriate N-terminal domain of receptor-binding proteins when engineering bacteriophages with altered host specificity.

## 1. Introduction

The escalating prevalence of multidrug-resistant (MDR) and extensively drug-resistant (XDR) bacteria represents a significant global health threat. These pathogens, resistant to most conventional antibiotics, are directly linked to prolonged illnesses, elevated mortality rates, and untreatable infections, requiring the urgent development of novel therapeutic strategies. Bacteriophages (phages), the natural enemies of bacteria, represent a promising alternative for the effective control of bacterial infections [1,2]. Due to their distinct mechanisms of action, antibiotics and phages can complement each other to overcome bacterial resistance [3,4,5].

Advances in genetic engineering and synthetic biology have enabled the design of synthetic phages with tailored properties based on natural phage scaffolds (reviewed in [6,7,8,9,10,11]). A particularly inspiring application of phage engineering is the reprogramming of host specificity by exchanging receptor-binding proteins (RBPs) between phages. This strategy has been successfully used to redirect phage specificity both within and across bacterial genera [12,13,14].

Phage RBPs often exhibit a modular architecture: a conserved N-terminal adapter domain anchoring the RBP to the phage tail and a highly variable C-terminal domain that determines receptor specificity [15,16]. This modularity is believed to facilitate rapid phage adaptation to new hosts through horizontal gene transfer and domain swapping [13,17].

Many podoviruses utilize a T7gp17-like adapter domain (Pfam PF03906) to attach RBPs to the tail [15,16,18,19]. In particular, tailspike proteins (tsp) of *Klebsiella*-infecting phages belonging to the *Przondovirus* and *Drulisvirus* genera contain this domain [20,21]. In addition to the type A tailspikes, which are directly attached to the tail using the T7gp17-like adapter domain, phages of these genera can also contain auxiliary type B tailspikes that are attached to a T4gp10-like branching domain of the type A tailspikes [15,20,22]. Although the T7gp17-like adapter domain is structurally conserved across many phages, its amino acid sequence is highly divergent. This sequence variation, even among phages of the same genus, can prevent the incorporation of heterologous RBPs into a recipient phage’s tail, complicating the engineering of phages with an altered host specificity [12]. A potential solution is the construction of chimeric RBPs, in which the N-terminal adapter domain from the recipient phage is fused to the receptor-binding domain from a donor phage, thereby ensuring proper tail assembly [13].

In this study, we applied this approach to reprogram the host specificity of two *Przondovirus* phages, KP192 and KP195, targeting *Klebsiella* strains of different capsular types. While first-generation synthetic phages KP192_tspA195 and KP195_tspAB192 with switched specificity were previously constructed through whole RBP gene replacement [14], we hypothesized that preserving the native N-terminal domain of the recipient phage’s type A tailspikes would enhance tail assembly efficiency. We therefore engineered second-generation phages KP192_tspN_192_A195 and KP195_tspN_195_AB192 featuring chimeric type A tailspikes with the specificity-determining C-terminal part of the donor phage’s tailspikes and the N-terminal domain of the recipient phage’s tailspikes.

The obtained results indicated that the N-terminal domain of the type A tailspikes serves not merely as a structural adapter but also modulates the receptor-binding and hydrolytic activities of the tailspikes. Consequently, its replacement can impair phage fitness. These findings underscore the importance of choosing an appropriate N-terminal domain of receptor-binding proteins when engineering bacteriophages with altered host specificity.

## 2. Results

### 2.1. Experimental Design and Construction of Synthetic Phages

The effect of the T7gp17-like adapter domain of type A tailspike proteins on phage properties was studied using phages KP192 and KP195 from the Collection of Extremophilic Microorganisms and Type Cultures (CEMTC) of the Institute of Chemical Biology and Fundamental Medicine SB RAS (ICBFM SB RAS), Novosibirsk. These phages exhibit a podoviral morphology, belong to the *Przondovirus* genus, and have approximately 60 nm capsids harboring genomes of 40,635 bp and 40,540 bp, respectively (GenBank accession numbers: NC_047968 and NC_047970). The KP192 virion contains six copies of the homotrimeric KL111-specific type A tailspike (tspA192) encoded by the *tspA192* gene and integrated into the phage tail, and an additional six copies of the homotrimeric K2-specific type B tailspike (tspB192), encoded by the *tspB192* gene and attached to the T4gp10-like branching domain of the type A tailspikes (Figure 1A–C and Table 1) [14,20,23]. Thus, this phage infects *Klebsiella* with K2 and KL111 capsular types. The KP195 virion contains only K64-specific type A tailspikes (tspA195) encoded by the *tspA195* gene and embedded directly into the tail (Figure 1A,C and Table 1), and therefore infects *Klebsiella* with a K64-type capsule. The tailspike proteins of phages KP192 and KP195 share >95% identity with those of the *Klebsiella* phages Kp9 and SH-Kp 152410, respectively [14,24,25].

The sequences of the N-terminal domains (the first 149 amino acid residues) of the tspA192 and tspA195 proteins differ significantly (68% identity), although the AlphaFold3-generated models have a similar fold, which is typical for other T7gp17-like NTDs (PDB ID: 7BOZ, 8DSP, 7EY9, and 7Y1C) (Figure 2A,B). Some of the amino acid substitutions are located at the interface formed by the NTD of the type A tailspike and the nozzle and gatekeeper proteins that form the phage tail (Figure 2C). Structural analysis of the tail model, performed using UCSF Chimera, revealed that none of these differences result in pronounced steric hindrances or clashes between amino acid residues. However, there is a possibility that differences in these regions can reduce the efficiency of incorporation of type A spikes of the KP192 phage into the tail of the KP195 phage and vice versa, thereby reducing the replication efficiency.

To investigate this effect, synthetic phages similar to the previously described KP192_tspA195 and KP195_tspAB192 [14], but featuring chimeric type A tailspikes, were constructed. The N-terminal domain (the first 149 amino acid residues) in such chimeric spikes corresponded to the tail of the recipient phage, and the receptor-binding domains were transferred from the RBP of the donor phage and provided the required capsular specificity (Figure 1A,C, Table 1). Following the previously established naming scheme, these phages were designated KP195_tspN_195_AB192 and KP192_tspN_192_A195, where the first part indicates the genomic scaffold, followed by an underscore and the name of the transplanted gene(s). It is worth noting that phages KP195_tspN_195_AB192 and KP192_tspN_192_A195 differed from the corresponding phages KP195_tspAB192 and KP192_tspA195 only in the region encoding the first 149 amino acid residues of the tailspike A proteins, and were otherwise identical. Therefore, a comparison of the properties of these phages allowed us to determine the effect of differences in the N-terminal domain of the type A tailspikes on phage properties.

Synthetic phage genomes were assembled via transformation-associated recombination (TAR) cloning in yeast (Figure 3A) [26,27,28,29]. Each genome was split into nine overlapping fragments (3–6 kbp) in relatively conserved regions. These fragments were PCR-amplified and combined with a part of the yeast centromeric plasmid pRSII415, following the previously described approach [12,13,14,30]. The detailed scheme of the assembly is shown in Figure 3B. To ensure that the mutation rate was acceptable and did not interfere with the formation of viable virions, the genomes of control synthetic phages KP192ctrl and KP195ctrl, identical to the parental phages KP192 and KP195, were assembled similarly. Following yeast transformation, individual colonies were screened using PCR. Yeast plasmid DNAs containing synthetic phage genomes were isolated from the positive clones after yeast cultivation and used for phage genome “rebooting”. Following phage amplification, the correct assembly of the synthetic phage genomes was confirmed using PCR verification and sequencing. The properties of the wild-type and the synthetic phages are summarized in Table 1.

### 2.2. “Rebooting” of the Synthetic Klebsiella Phage Genomes

Since *K. pneumoniae* strains often demonstrate low electrocompetence [31], synthetic phages were produced by transformation of *E. coli* cells as an intermediate host, similarly to the method described previously [12,13,14]. *E. coli* cell extracts containing phage particles were used to infect *K. pneumoniae* strains with a suitable K-type. The *K. pneumoniae* strain CEMTC-2274 (hereinafter A_KL111_) with the capsular type KL111 was chosen for propagation of KP195_tspN_195_AB192 and KP192ctrl phages, since both had KL111-specific tailspikes. The *K. pneumoniae* strain CEMTC-2337 (hereinafter E_K64_) was used for K64-specific phages KP192_tspN_192_A195 and KP195ctrl for similar reasons.

The genomes of the control synthetic phages, KP192ctrl and KP195ctrl, were successfully “rebooted”, resulting in the formation of numerous plaques on a *Klebsiella* lawn (Figure 3C). The shape and size of plaques formed by the corresponding control and wild-type phages were the same. This indicated a high efficiency of assembly and “rebooting” of the genomes of synthetic *Klebsiella* phages using *E. coli* as an intermediate host. However, phages KP195_tspN_195_AB192 and KP192_tspN_192_A195 could not be “rebooted” using strains A_KL111_ and E_K64_, respectively. It has been previously shown that replacing the phage KP192 genomic scaffold with the phage KP195 genomic scaffold (and vice versa) resulted in a decrease in the replication efficiency of chimeric phages KP195_tspAB192 and KP192_tspA195 on strains A_KL111_ and E_K64_, respectively, compared to wild-type phages KP192 and KP195 [14]. Therefore, strains CEMTC-2291 (hereinafter B_K2_) and CEMTC-11039 (hereinafter H_K64_), on which phages KP195_tspAB192 and KP192_tspA195 replicated more efficiently [14], were also used to “reboot” the genomes of phages KP195_tspN_195_AB192 and KP192_tspN_192_A195. Finally, the genomes of phages KP195_tspN_195_AB192 and KP192_tspN_192_A195 were successfully “rebooted” (Figure 3C), resulting in the formation of phage particles. The phages were eluted from plaques and used in further experiments.

### 2.3. N-Terminal Domains of Type A Tailspikes Alter Phage Replication Efficiency via Different Mechanisms

To test whether the N-terminal domain sequence can impact phage properties, the infectious characteristics of the KP195_tspN_195_AB192 and KP192_tspN_192_A195 phages were compared to those of the KP195_tspAB192 and KP192_tspA195 synthetic phages, described previously [14] (Figure 1A,C, Table 1). The efficiency of plating (EOP) and the efficiency of planktonic cell lysis were studied using *Klebsiella* strains with suitable K-types. In order to correctly compare different phages, equalization of the phage particle concentration in the phage suspensions was performed using protein electrophoresis followed by densitometry (Appendix A), as described and validated previously [14].

The infectious properties of phages KP195_tspAB192 and KP195_tspN_195_AB192 were studied on strains A_KL111_ (since the KL111 capsular type is rare, this strain was the only available KL111 strain), B_K2_, and two additional K2 strains CEMTC-2573 and CEMTC-3533 (hereinafter C_K2_ and D_K2_, respectively). It was found that infectious titers differed by ≤1.5 orders of magnitude on K2 strains, and the appearance of the plaques was the same (Figure 4A,B and Appendix A). These results were confirmed using five additional K2 strains (Appendix A). However, a significant difference was observed on the A_KL111_ strain with the KL111-type capsule: phage KP195_tspN_195_AB192 formed very small plaques, and the infectious titer of the phage sample was 4 orders of magnitude lower than that of phage KP195_tspAB192 (Figure 4A,B). The adsorption efficiency of the KP195_tspN_195_AB192 phage also depended on the K-type of the *Klebsiella* strain (Figure 4C). On the B_K2_ strain, both synthetic phages KP195_tspAB192 and KP195_tspN_195_AB192, as well as the parental phage KP192, were adsorbed with the same efficiency. However, on the A_KL111_ strain, the adsorption efficiency of the KP195_tspN_195_AB192 phage was significantly lower. In addition, the capsule hydrolytic (depolymerase) activity of UV-inactivated phages was studied. No differences were found on the K2-type strains, but on the KL111-type strain, the inactivated KP195_tspN_195_AB192 phage showed weaker activity (Figure 4D). These results show a clear pattern: the KP195_tspAB192 and KP195_tspN_195_AB192 phages differed only in the NTD of the KL111-specific type A tailspikes, and the most striking differences in the properties of these phages were observed exactly on the KL111-type strain.

The efficiency of plating for the second pair of phages (KP192_tspA195 and KP192_tspN_192_A195) could not be compared quantitatively: it was not possible to obtain a suspension of the KP192_tspN_192_A195 phage with a concentration high enough to determine the concentration of phage particles by electrophoresis. This was due to the fact that the replication efficiency of the KP192_tspN_192_A195 phage was low: this phage formed small turbid plaques even on those strains on which the KP192_tspA195 phage formed large plaques (Figure 4E and Appendix A). In addition, the KP192_tspN_192_A195 phage did not cause lysis of planktonic cultures of strains H_K64_ and CEMTC-11041, while the KP192_tspA195 phage effectively lysed them. However, the differences in the properties of these phages clearly indicated that the chimeric tailspike protein tspN_192_A195 reduced the reproduction efficiency of the KP192_tspN_192_A195 phage compared to that of the KP192_tspA195 phage. This is similar to the reduction in the reproduction efficiency of phage KP195_tspN_195_AB192 compared to that of phage KP195_tspAB192 on strain A_KL111_ due to the chimeric tailspike tspN_195_A192.

Next, the efficiency of lysis of planktonic cultures of strains A_KL111_ and B_K2_ by phages KP195_tspAB192 and KP195_tspN_195_AB192 was studied. It was found that KP195_tspN_195_AB192, unlike KP195_tspAB192, did not cause lysis of the A_KL111_ culture (Figure 4F), which is consistent with its lower efficiency of plating on this strain. However, this phage resulted in significantly more effective lysis of the B_K2_ culture (Figure 4F). To study this phenomenon, one-step growth experiments were performed, allowing us to determine the burst size and latent period for these phages. The latent period was approximately 10 min for both phages on both strains. However, the burst size differed substantially. When infecting strain A_KL111_, KP195_tspN_195_AB192 produced approximately 3 phages per cell, whereas KP195_tspAB192 yielded approximately 25 phages per cell (Figure 4G). Conversely, on strain B_K2_, the burst size of KP195_tspN_195_AB192 reached approximately 80 phages per cell, compared to approximately 17 phages per cell for KP195_tspAB192. These data agreed well with the observed lytic efficiency profiles and the size and morphology of the phage plaques.

### 2.4. Sequence Conservation of N-Terminal Domains of tspA192 and tspA195 Tailspikes

To determine whether there is a correlation between the sequence of the enzymatic part of type A tailspikes and the sequence of the NTD of these proteins, a bioinformatics search for tailspike proteins similar to tspA192 and tspA195 was performed using the NCBI Protein BLAST (https://blast.ncbi.nlm.nih.gov/Blast.cgi (accessed on 10 July 2025)) against the non-redundant protein sequences (nr) database. Sequences with >75% identity and >80% query coverage were selected. All 18 proteins closely related to tspA192 had a highly conserved NTD sequence similar to that of the N_192_ domain (Table 2). All 84 proteins closely related to tspA195 also had a conserved NTD sequence similar to that of the N_195_ domain (Table 2). Meanwhile, the mean distance between the N_192_-like group and the N_195_-like group was 0.403, indicating that these two groups differed substantially. In addition, a phylogenetic analysis of the N-terminal part of the studied sequences formed the same groups as an analysis of their enzymatic part (Figure 5). Thus, a clear correlation was demonstrated between the sequences of the enzymatic part of type A tailspikes and the sequences of their NTD in natural phages. No genes were found that encode a hybrid tsp protein containing an NTD from one group and enzymatic/receptor-binding domains from another group. This apparently reflects the fact that such hybrid tsp proteins reduced the efficiency of phage reproduction and were eliminated during evolution.

## 3. Discussion

Engineering of synthetic phages with altered specificity requires correct docking of RBPs of a donor phage with the virion of a recipient phage. Several options are possible: (1) to transfer the entire RBP gene, including the part encoding the adapter domain; (2) to assemble the gene of a chimeric RBP containing the adapter domain from the recipient phage; and (3) to transfer all the genes encoding the tail proteins (i.e., the RBP(s), the gatekeeper and the nozzle proteins) from the donor phage. The first option is suitable only for closely related phages, since only in this case can the N-terminal adapter domain of the RBP of one phage seamlessly dock with the tail of another phage. The second option ensures that the chimeric RBP can integrate into the tail due to the presence of the appropriate adapter domain. Therefore, it allows transplantation of RBP between distant phages, including phages of different morphological types. However, the transferred RBP does not always function properly in the context of a “foreign” tail [12,13]. The third option is sometimes the only working solution, for example, when constructing *Klebsiella*-specific phages based on a genomic scaffold of phage T7 [12]. However, this option is only applicable to phages with the same tail morphology and requires smooth docking of the transferred tail of one phage with the portal of another phage.

In this study, we compared the first two approaches using phages belonging to the *Przondovirus* genus and infecting *K. pneumoniae*. Surprisingly, the first approach (replacement of the entire gene encoding a type A tailspike) turned out to be more effective and robust, despite a mismatch between the N-terminal domains of the type A tailspike and the rest of the tail proteins. The efficiency of replication of the KP195_tspAB192 and KP192_tspA195 phages using *Klebsiella* strains with the capsular types KL111 or K64, respectively, was substantially higher compared to that of the second-generation phages KP195_tspN_195_AB192 and KP192_tspN_192_A195 containing chimeric tspA proteins. Since the genomes of the first- and second-generation phages differed only in the region encoding the N-terminal domain of the type A tailspike, it can be concluded that either the N-terminal domain itself or the corresponding region of the genome significantly influenced the efficiency of phage reproduction.

Several mechanisms can be proposed to explain the observed results. One of the most obvious reasons is the misfolding of the type A spikes, resulting in the formation of phages with defective tails. However, since type B spikes are attached to the phage tail only through the T4gp10-like branching domain of the type A spikes [13,15,20,23], misfolding or the absence of a significant number of type A spikes would lead to more radical differences in the properties (efficiency of plating, size and transparency of plaques, adsorption rate, and depolymerase activity of UV-inactivated phages) of the KP195_tspN_195_AB192 and KP195_tspAB192 phages on K2 strains, comparable to the differences on the A_KL111_ strain. This was not observed.

Another hypothesis is that the N-terminal domain of the type A tailspikes is directly involved in the binding (and, possibly, hydrolysis) of the capsular polysaccharide, thereby enhancing the depolymerase activity of the enzymatic/binding domains of the type A tailspikes. This hypothesis is supported by the fact that the adsorption efficiency of the KP195_tspN_195_AB192 phage on the cells of the A_KL111_ strain, as well as the depolymerase activity of the UV-inactivated KP195_tspN_195_AB192 phage against this strain, was reduced compared to that of the KP195_tspAB192 phage. Apparently, the N_195_ domain located in the chimeric type A spikes of the KP195_tspN_195_AB192 phage was unable to enhance the binding of these spikes to the capsule of the A_KL111_ strain. This hypothesis is also supported by the fact that the gatekeeper and nozzle proteins of the phage tail can exhibit polysaccharide-hydrolytic activity [32,33,34,35], although this activity is auxiliary for these proteins. Our study provides the first evidence that the NTD of the phage tailspikes, generally considered as an adapter domain, can modulate receptor-binding and degrading activity. It is possible that this additional receptor-binding site proposed above is formed only upon contact of the N-terminal domain of the tailspikes with other tail proteins (gatekeeper and nozzle) and therefore cannot be detected by studying individual tailspike proteins.

Furthermore, potential steric hindrance between the N-terminal domain of the type A tailspikes and other tail proteins was investigated. Structural analysis of the N_192_ and N_195_ domains did not reveal amino acid residues likely to cause unfavorable interactions or steric clashes with adjacent gatekeeper or nozzle proteins. In addition, one-step growth experiments revealed that the burst size of KP195_tspAB192 was higher than that of KP195_tspN_195_AB192 on strain A_KL111_, while the opposite was true on strain B_K2_. The burst size depends on the efficiency of virion assembly, among other factors. Taking all these data into account, there is currently no evidence that the N-terminal domain of the KP192 phage type A tailspike reduced the incorporation efficiency of these tailspikes into the KP195 phage tail, or vice versa.

Phage replication efficiency is the product of efficiencies of all successive infection stages: adsorption, capsule degradation, DNA ejection, expression of phage genes, genome replication, virion assembly, and host cell lysis. In our one-step growth experiments, the burst size represents the ratio of the number of phage particles released upon host cell lysis to the number of lysed cells and therefore is independent of the efficiencies of adsorption, capsule degradation, DNA ejection, and cell lysis. Therefore, the observed 5- to 7-fold difference in the burst size indicated that the sequence of the N-terminal adapter domain of type A tailspikes can also affect the efficiency of the intracellular stages of the infection cycle. This effect may be attributable to one of the host anti-phage defense systems, though it is surprising that a 440 bp DNA fragment encoding the N-terminal domain of the tailspike protein could trigger such a response.

Our findings on the role of the N-terminal domain are not entirely consistent with a previous study, which reported no functional differences in the properties of related *Przondovirus* phages harboring native versus chimeric type A tailspikes [13]. This discrepancy may be attributed to two factors: first, the prior study did not include a quantitative comparison of replication efficiencies. Second, the sequence divergence between the type A tailspike NTDs of the K11 and KP32 phages used in that study (11%) was considerably lower than the 32% divergence in our KP192/KP195 system. Therefore, the difference in the properties of the phages in our study could be more pronounced.

It is believed that the adapter/anchor domains and receptor-binding domains of RBPs of bacteriophages are independent modules with clearly distinct functions and that phages can exchange receptor-binding domains via horizontal gene transfer [13,17]. Our results demonstrated that the T7gp17-like N-terminal domains of phage RBPs are not merely structural adapters but can also enhance the receptor-binding and hydrolytic activities of the tailspikes. Therefore, replacing the NTD of an RBP with a related one can lead to significant changes in the efficiency of receptor binding and hydrolysis.

A bioinformatics analysis of tspA192-like and tspA195-like tailspikes also indicated a high correlation between the sequences of the NTD and the enzymatic/binding domain of the type A spikes in natural phages. This co-evolution suggests that the functions of these domains are interconnected, and unfavorable combinations reduced phage fitness and were eliminated during evolution.

In conclusion, the study demonstrated that the sequence of the N-terminal domain of the type A tailspikes of the *Przondovirus* phages can exert a complex influence on the infectious properties of the phage, affecting both pre- and post-ejection stages of the infection cycle. To improve the efficiency of the synthetic phage design and the predictability of their properties, the above factors should be taken into account when choosing an appropriate strategy for the transfer of receptor-binding proteins between phages.

## 4. Materials and Methods

### 4.1. Phages, Bacterial, and Yeast Strains

The wild-type *Klebsiella* phages KP192 and KP195 (GenBank accession numbers NC_047968 and NC_047970, respectively) were obtained from the Collection of Extremophilic Microorganisms and Type Cultures (CEMTC) at the Institute of Chemical Biology and Fundamental Medicine, Siberian Branch of the Russian Academy of Sciences (ICBFM SB RAS), Novosibirsk. Two synthetic phages, KP195_tspAB192 and KP192_tspA195, were used from a previous study [14]. *Saccharomyces cerevisiae* strain BY4741 (ATCC 4040002) was employed for transformation-associated recombination (TAR) cloning. *Escherichia coli* TOP10 (Thermo Fisher Scientific, Waltham, MA, USA) was used for the “rebooting” of phage genomes. The following *Klebsiella pneumoniae* strains were used for phage amplification and characterization: KL111-type strain CEMTC-2274 (designated A_KL111_); K2-type strains CEMTC-2291 (B_K2_), CEMTC-2573 (C_K2_), CEMTC-3533 (D_K2_), CEMTC-11061, CEMTC-11062, CEMTC-11063, CEMTC-11064, and CEMTC-11066; and K64-type strains CEMTC-2337 (E_K64_), CEMTC-11034, CEMTC-11036, CEMTC-11037, CEMTC-11038, CEMTC-11039 (H_K64_), and CEMTC-11041. The K-locus types of these strains were previously determined either by *wzi* gene sequencing (GenBank accession numbers: MN371474, MN371475, MN371483, MN371512, and MN371476) or by *wzy* allele-specific PCR [14,36].

### 4.2. Culturing Conditions

Bacterial cultures were grown at 37 °C on Lysogeny Broth (LB) agar plates or in liquid LB medium with shaking at 180 rpm. *S. cerevisiae* suspension cultures were incubated at 27–30 °C with shaking at 180 rpm in either rich YPD medium (1% yeast extract, 2% peptone, 2% dextrose) or synthetic selective YNB-Leu medium, composed of 0.67% Yeast Nitrogen Base with ammonium sulfate (BD Biosciences), 2% dextrose, and 0.069% Complete Supplement Mixture lacking leucine (CSM-Leu; MP Biomedicals, Santa Ana, CA, USA).

### 4.3. Preparation of DNA Fragments for Assembly of Phage Genomes

Overlapping fragments of the phage genomes were amplified by PCR with Phusion High-Fidelity DNA Polymerase (Thermo Fisher Scientific, Waltham, MA, USA) using the primers listed in Appendix A and following the manufacturer’s protocol. A small aliquot (0.1–0.2 μL) of a phage suspension at a titer of 10^8^–10^10^ PFU/mL was used as a template. The vector fragment was amplified from the yeast centromeric plasmid pRSII-415 (Addgene plasmid #35454) using the primers pRSII415_192/5_genome_dir and pRSII415_192/5_genome_rev (Appendix A). The resulting vector fragment was used to assemble both KP192- and KP195-based phage genomes. All the fragments were purified with the GeneJET Gel Extraction Kit (Thermo Fisher Scientific, Waltham, MA, USA), and DNA concentration values were determined using a NanoDrop One spectrophotometer (Thermo Fisher Scientific, Waltham, MA, USA).

### 4.4. Phage Genome Assembly in Yeast

Transformation-associated recombination (TAR) cloning in yeast was used for phage genome assembly [26,29,37]. Competent cells of the *S. cerevisiae* strain BY4741 were prepared according to previously described methods [38,39]. For the transformation, approximately 300 ng of each appropriate phage genome fragment (Appendix A) and 300 ng of the vector backbone fragment were combined with 240 μL of 50% PEG-3350, 36 μL of 1 M lithium acetate, and 25 μL of denatured salmon sperm DNA (2 mg/mL). The final reaction volume was adjusted to 360 μL. Subsequently, approximately 10^8^ freshly prepared competent yeast cells were added to the mixture. The cell suspension was heat-shocked at 42 °C for 30–60 min. Following incubation, the cells were pelleted by centrifugation at 12,000× *g* for 30 s and resuspended in 200 μL of sterile water. The transformed yeast cells were plated on YNB-Leu agar plates and incubated at room temperature for 3–4 days to allow for colony formation.

### 4.5. Isolation of a Yeast Centromeric Plasmid Harboring the Phage Genome

Individual yeast colonies were inoculated separately into YNB-Leu medium and grown at 30 °C with shaking (180 rpm) until the optical density OD_600_ reached 6–9. Total DNA, including the centromeric plasmid pRSII-415 harboring the phage genome, was extracted from the yeast cells using a previously described method [12,13].

### 4.6. Phage Genome “Rebooting”

Phage genomes were rebooted using *E. coli* as an initial phage propagation host, as previously described [12,13]. Briefly, an aliquot of the total yeast DNA preparation, containing the cloned phage genome, was used to transform electrocompetent *E. coli* TOP10 cells (Thermo Fisher Scientific, Waltham, MA, USA) via electroporation. Following electroporation, the cells were recovered in 1 mL of SOC medium and were grown at 37 °C for 3 h with shaking. To induce phage release, the cells were lysed by the addition of 50 μL of chloroform, followed by vigorous vortexing and centrifugation at 12,000× *g* for 1 min. The resulting supernatant was mixed with 0.5 mL of an exponentially growing culture of the appropriate *K. pneumoniae* strain and 4 mL of molten top agar (0.8% *w*/*v*). The mixture was poured onto LB agar plates. Phage plaques were observed following incubation at 37 °C for 3–16 h.

### 4.7. Verification of Genome Assembly Accuracy

Four pairs of phage-specific primers (Appendix A) were used in PCR to verify that the synthetic phage samples contained chimeric genomes (the genomic scaffold from one phage and the transferred tailspike gene from another phage), as previously described [14].

A 700 bp region, starting at nucleotide position 33,024 and covering a junction between the genes encoding the internal virion protein D and the type A tailspike (including the entire N-terminal domain-encoding sequence), was verified in the genomes of phages KP192_tspA195 and KP192_tspN_192_A195. The region was amplified by PCR using primers pt7_ivpD192/5_seq_dir and pt8_tsp195_seq_v2_rev (Appendix A) and sequenced by the Sanger method with a BigDye Terminator v3.1 kit (Thermo Fisher Scientific, Waltham, MA, USA) according to the manufacturer’s instructions.

Complete genome sequencing of phages KP195_tspAB192 and KP195_tspN_195_AB192 was performed as described previously [40]. Briefly, phage genomic DNA was fragmented using a Covaris Ultrasonicator (Covaris, Woburn, MA, USA), and DNA libraries were constructed using the NEB Next Ultra II DNA Library Prep Kit for Illumina (both from New England BioLab, Ipswich, MA, USA). Paired-end sequencing was performed on an Illumina MiSeq sequencer using a v.2 reagent kit (2 × 250 base reads) (Illumina Inc., San Diego, CA, USA). The phage genomes were assembled de novo using SPAdes Genome Assembler v. 3.15.4 [41].

### 4.8. Phage Propagation and Purification

Phages were propagated by infecting 50 mL of an exponentially growing culture (OD_600_ = 0.4–0.7) of the appropriate *K. pneumoniae* host strain at a multiplicity of infection (MOI, i.e., the ratio of phage to bacterium, calculated based on the infectious titer of the phage sample) of 0.01 (1 PFU of phage per 100 cells). The following phage-host pairs were used: KP192ctrl and KP195_tspAB192 with strain A_KL111_; KP195_tspN_195_AB192 with strain B_K2_; KP195crtl with strain E_K64_; and KP192_tspA195 with strain H_K64_. Infected cultures were incubated with shaking at 37 °C until complete lysis was observed. Cellular debris was removed by centrifugation, and phages were purified from the supernatant by precipitation with polyethylene glycol 6000 (PEG-6000), as described previously [42,43]. The phage pellet was resuspended in 800 μL of SM buffer (10 mM NaCl, 10 mM MgCl_2_, 50 mM Tris-HCl, pH 7.5, 0.05% NaN_3_) and stored at 4 °C.

### 4.9. Determination of Infectious Titer

Two types of phage titers were used in this study: the infectious titer (specific for a particular *Klebsiella* strain) and the pseudo-physical titer (independent of *Klebsiella* strain), introduced previously [14]. The infectious titer (PFU/mL) of phage samples was determined by the double-agar overlay plaque assay. An appropriate indicator *Klebsiella* strain was grown in LB medium at 37 °C with shaking (180 rpm) to mid-exponential phase (OD_600_ = 0.5–0.6). A 0.5 mL aliquot of the bacterial culture was mixed with 4 mL of molten soft agar (0.8%) and overlaid onto an LB agar plate. Ten-fold serial dilutions (10^−1^ to 10^−8^) of each phage sample were prepared in LB medium. Aliquots (6 µL) of each dilution were applied onto the prepared bacterial lawns. Plates were incubated overnight at 37 °C, and plaques were counted to calculate the titer. All assays were performed in triplicate. Data analysis was performed using Microsoft Excel 2010.

### 4.10. Equalization of Phage Particle Concentration in Phage Samples

To enable a direct and fair comparison between phages with different efficiencies of plating, the concentrations of phage virions in phage stocks were normalized (or equalized) using protein electrophoresis followed by densitometry, as validated previously [14]. A pseudo-physical phage titer (titer_PP_), measured in “protein concentration-linked units” per milliliter (PCLU/mL), was used to describe phage particle concentration. The infectious titer of phage KP192 determined on the bacterial lawn formed by an exponentially growing culture of strain A_KL111_ was chosen as a reference for determining the PCLU units. Thus, if a sample of the KP192 phage had an infectious titer of 10^11^ PFU/mL when analyzed using strain A_KL111_, then it was considered to contain exactly 10^11^ PCLU in 1 mL.

To determine the pseudo-physical titer of a test phage, two-fold serial dilutions of the sample were analyzed by SDS-PAGE together with an aliquot of the reference sample containing 3 × 10^8^ PCLU of purified phage KP192. Following electrophoresis, the gels (12% *w*/*v*) were stained using Coomassie G-250 (Appendix A). The band intensities corresponding to the major capsid protein (MCP; ~37 kDa) were quantified using Image-Lab 6.0 software (Bio-Rad). The dilution of the test sample whose MCP band intensity most closely matched that of the reference sample was used to calculate the titer_PP_ using the following equation:titer_PP_ = OD_test_ × DF × (3 × 10^8^ PCLU)/(OD_reference_ × V_test_)(1)
where OD_test_ and OD_reference_ are the densitometry values of the major capsid protein bands for the test and reference samples, respectively, V_test_ is the volume of the test sample aliquot analyzed by SDS-PAGE, and DF is the dilution factor of the test sample. The estimated error rate for this method is 20–30%, which is acceptable for the purposes of this study.

### 4.11. Determination of Phage Adsorption Efficiency

A test strain of *K. pneumoniae*, used for phage adsorption, and an indicator strain of *K. pneumoniae*, needed to count plaques formed by unbound phage particles, were used in each adsorption experiment. Strain A_KL111_ was used as the indicator strain for phage KP192, while strain B_K2_ was used as the indicator for the KP195_tspAB192 and KP195_tspN_195_AB192 phages since these phage-strain combinations provided efficient formation of large plaques. When used for phage adsorption, *K. pneumoniae* strains A_KL111_ and B_K2_ were cultivated at 37 °C until the OD_600_ reached 0.2. Bacterial cultures used as indicator strains (i.e., for plaque formation) were grown under the same conditions to the OD_600_ of 0.4–0.6. An aliquot of the test phage containing 5 × 10^6^ PCLU in 10 μL was mixed with 100 μL of the adsorption strain suspension (10^7^ CFU) or with 100 μL of LB medium (control experiment). Following incubation at 37 °C for 7 min, all samples were centrifuged at 12,000× *g* for 30 s to settle down the cells and adsorbed phages.

A 50 μL aliquot of the supernatant containing unadsorbed phages was mixed with 350 μL of PBS and 20 μL of chloroform. Following centrifugation at 12,000× *g* for 1 min, the supernatants were serially ten-fold diluted in LB medium. Then, 100 μL of each dilution was mixed with 500 μL of the indicator strain culture and 3.5 mL of molten top agar (0.8%), and the mixture was overlaid onto an LB agar plate. After incubation at 37 °C for 3–16 h, phage plaques were counted and the adsorption efficiency was calculated according to the following equation:adsorption efficiency (%) = [1 − (N_free_/N_control_) ] × 100%(2)
where N_control_ is the plaque count from the control sample (phage mixed with LB), and N_free_ is the plaque count on the experimental plate. The experiments were performed in triplicate, and the mean values and standard deviations (SD) were calculated.

### 4.12. Determination of Depolymerase Activity of UV-Inactivated Phages

Twenty microliters of phage suspensions (10^10^–10^11^ PCLU/mL) were irradiated with hard ultraviolet (UV) light under a low-pressure mercury-vapor discharge lamp for 90 min to inactivate the phages. To assay depolymerase activity, bacterial lawns were prepared by mixing 100 μL of an overnight culture of *K. pneumoniae* with 3.5 mL of molten top agar (0.8%) and overlaying the mixture onto LB agar plates. Serial ten-fold dilutions of the UV-inactivated phages in LB medium were applied to the top layer of agar (6 μL per spot). The plates were incubated at 37 °C for 3–5 h and examined for the formation of translucent zones indicating depolymerase activity.

### 4.13. Bacterial Killing Assay

An appropriate *K. pneumoniae* strain was cultivated at 37 °C with shaking at 180 rpm until the OD_600_ reached 0.5. A test phage aliquot containing 10^8^ PCLU was added to 5 mL of the bacterial culture containing approximately 10^9^ CFU (MOI = 0.1). After a 30 min adsorption period at 37 °C without shaking, the culture was incubated with shaking (180 rpm) at 37 °C. Starting from the moment of mixing the cells with the phage, 100 μL aliquots were collected, and the bacterial titer in each aliquot was determined by plating ten-fold serial dilutions onto LB agar plates. The experiments were performed in triplicate for each *Klebsiella* strain.

### 4.14. One-Step Growth Assay

An appropriate *K. pneumoniae* strain (test strain, e.g., A_KL111_ or B_K2_) was cultivated at 37 °C with shaking at 180 rpm until the OD_600_ reached 0.4–0.6. Phages were mixed with the test strain culture at an MOI of 0.02 (1 PCLU per 50 CFU). Following incubation at 37 °C for 5 min, the cells were centrifuged at 8000× *g* for 2 min to remove unbound phages. The cell pellet was suspended in fresh LB medium, and a fraction of the suspended cells was transferred to an incubation flask containing 10 mL of LB medium. The volume of the fraction was chosen to achieve a suitable concentration of infected cells for accurate plaque counting. To determine the number of infected cells that were able to undergo lysis, a 100 μL aliquot of the cell suspension from the incubation flask was immediately mixed with 500 μL of an exponential-phase B_K2_ culture (indicator strain, OD_600_ = 0.4–0.6) and 3.5 mL of molten top agar, and the mixture was overlaid onto an LB agar plate (“infected cells count” plate). Aliquots of the cell suspension were collected from the incubation flask every 5 min for 70 min and kept on ice. Following centrifugation at 12,000× *g* for 1 min at 4 °C, the aliquots were analyzed in the same way as the aliquot used for the “infected cells count” plate. After incubation at 37 °C for 3–16 h, phage plaques were counted and the burst size was calculated according to the following equation:Burst size = N_progeny_ × DF/N_inf. cells_(3)
where N_progeny_ is the number of plaques formed on a plate corresponding to a plateau in the growth curve (approximately 25 min after infection), DF is the dilution factor of the aliquot assayed, and N_inf. cells_ is the plaque count from the “infected cells count” plate.

### 4.15. Protein Structure Modeling and Visualization

The three-dimensional structures were predicted using AlphaFold3 [44], yielding high-confidence structures (pLDDT > 70) for most of the regions (Appendix A). For the branching domain of the type A tailspike, the homologous domain from phage VLCpiA3a (GenBank: UVX29715; residues 188–274) was selected over that of tspA192 (83% amino acid identity) due to its superior performance in modeling. The VLCpiA3a domain yielded a more plausible trimer model with tighter packing and a higher interface predicted TM-score (ipTM) of 0.63, compared to 0.17 for the tspA192 domain. Structural alignment, root-mean-square deviation (RMSD) calculation, model assembly, and visualization were performed using UCSF Chimera v. 1.13. [45].

### 4.16. Bioinformatics Analysis

Sequences of the C-terminal part (excluding the 149 N-terminal aa residues) of the tspA192 or tspA195 proteins were used to perform a BLAST (blastp algorithm using default parameters) (https://blast.ncbi.nlm.nih.gov/Blast.cgi (accessed on 10 July 2025)) search using the NCBI GenBank non-redundant protein sequences (nr) database. Sequences with ≤75% identity and/or ≤80% query coverage were removed. Each of the resulting sequences was split into N-terminal (the first 149 aa residues) and C-terminal regions. Phylogenetic analysis was conducted using MEGA v.7.0.26 [46]. Neighbor-joining trees with 500 bootstrap replicates were constructed separately for the N-terminal domain (NTD) and C-terminal (Cterm) sequence groups. The genetic distances between the NTDs of tspA192-like and tspA195-like proteins were estimated using the Dayhoff model, calculating the overall mean distance within each group and between the groups.

### 4.17. Quantification and Statistical Analysis

Data are presented as the mean ± standard deviation (SD). Statistical analyses were performed using GraphPad Prism 8.0.1 (GraphPad Software, USA). Where applicable, statistical significance was assessed by one-way analysis of variance (ANOVA) followed by Tukey’s test for multiple comparisons.

## Figures and Tables

**Figure 1 ijms-26-11297-f001:**
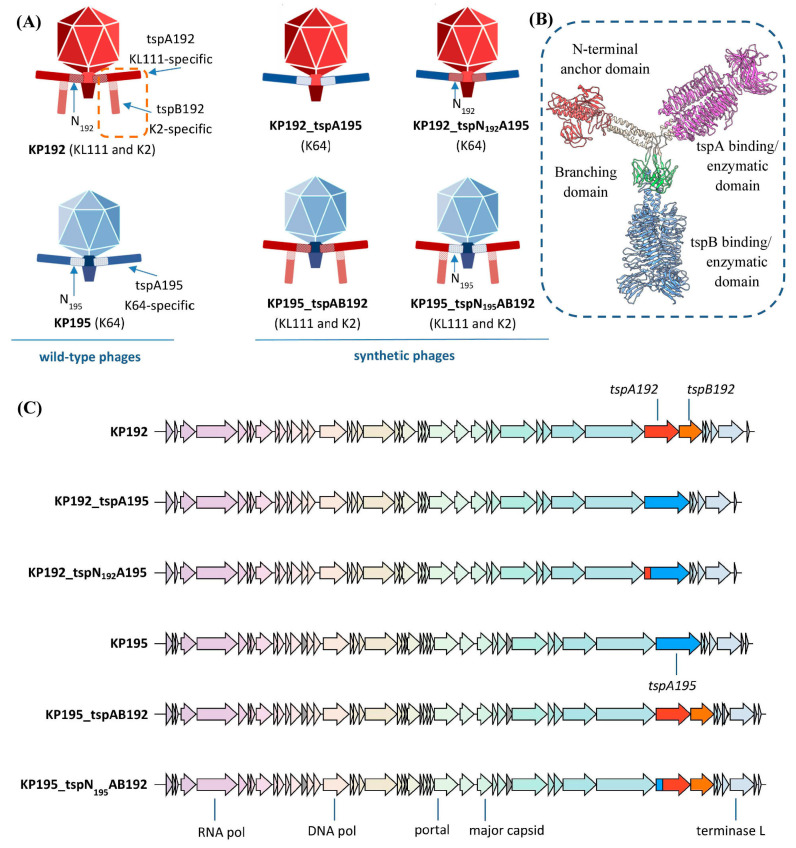
Bacteriophages KP192 and KP195 and the derived synthetic phages. (**A**) Schematic representation of the wild-type and synthetic phages. The capsular specificity of the synthetic phages is indicated in parentheses. “N_192_” and “N_195_” denote the N-terminal adapter domains of the tspA192 and tspA195 proteins, respectively. (**B**) A model of the homotrimeric type A tailspike with an attached homotrimer of type B tailspike (shown in light blue). The NTD of tailspike A is shown in red, the enzymatic domain is in magenta. The branching domain of spike A is shown in green. Parts of the model were predicted with high confidence using AlphaFold3 (https://alphafoldserver.com (accessed on 10 September 2025)). The final model was built and visualized using UCSF Chimera v.1.13.1. (**C**) Genomic maps of the bacteriophages. “RNA pol”—RNA polymerase gene; “DNA pol”—DNA polymerase gene; “terminase L”—terminase large subunit gene; “tspA” and “tspB”—genes of tailspike proteins A and B. The image was prepared using the clinker tool hosted at CAGECAT server (https://cagecat.bioinformatics.nl/tools/clinker (accessed on 10 September 2025)).

**Figure 2 ijms-26-11297-f002:**
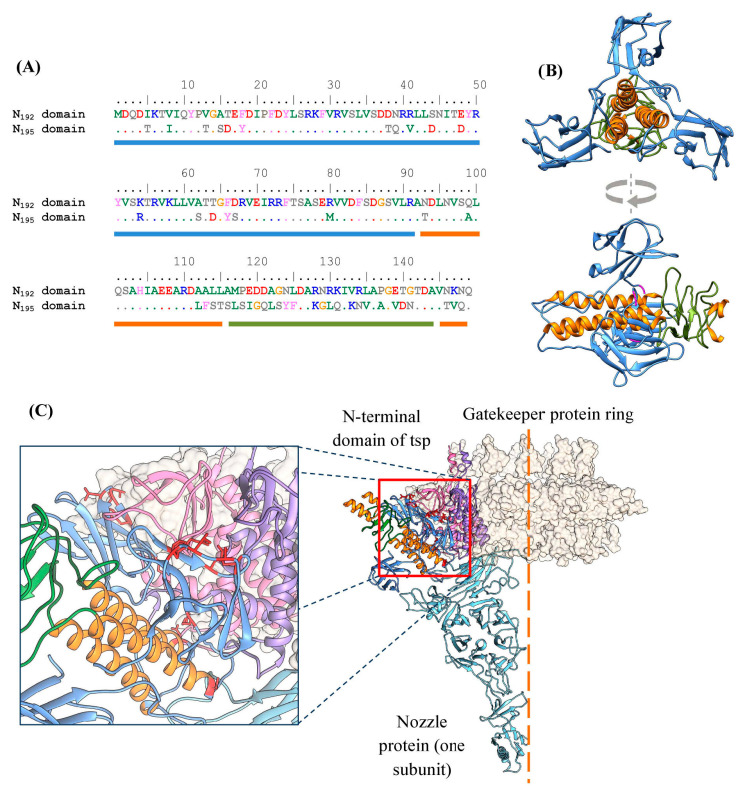
Differences between bacteriophages KP192 and KP195 and the derived synthetic phages. (**A**) Alignment of the N-terminal domain sequences of tspA192 and tspA195; colored bars below the alignment correspond to the domains shown in panel B. (**B**) Ribbon representation of the N-terminal domain of a tailspike protein homotrimer (based on the Kp9 phage tail structure, PDB ID: 7Y1C). (**C**) The interface between the N-terminal domain of a tailspike homotrimer (colored same as in (**B**)) and a ring formed by the gatekeeper protein (two interacting subunits out of twelve are colored in pink and violet, others are shown as a beige surface). One out of six subunits of the nozzle protein is shown as a sky-blue ribbon. Contact-forming residues that differ between the N_192_ and N_195_ domains are shown in red. The six-fold symmetry axis of the phage tail is shown as a dashed orange line. The images were prepared using UCSF Chimera 1.13.1.

**Figure 3 ijms-26-11297-f003:**
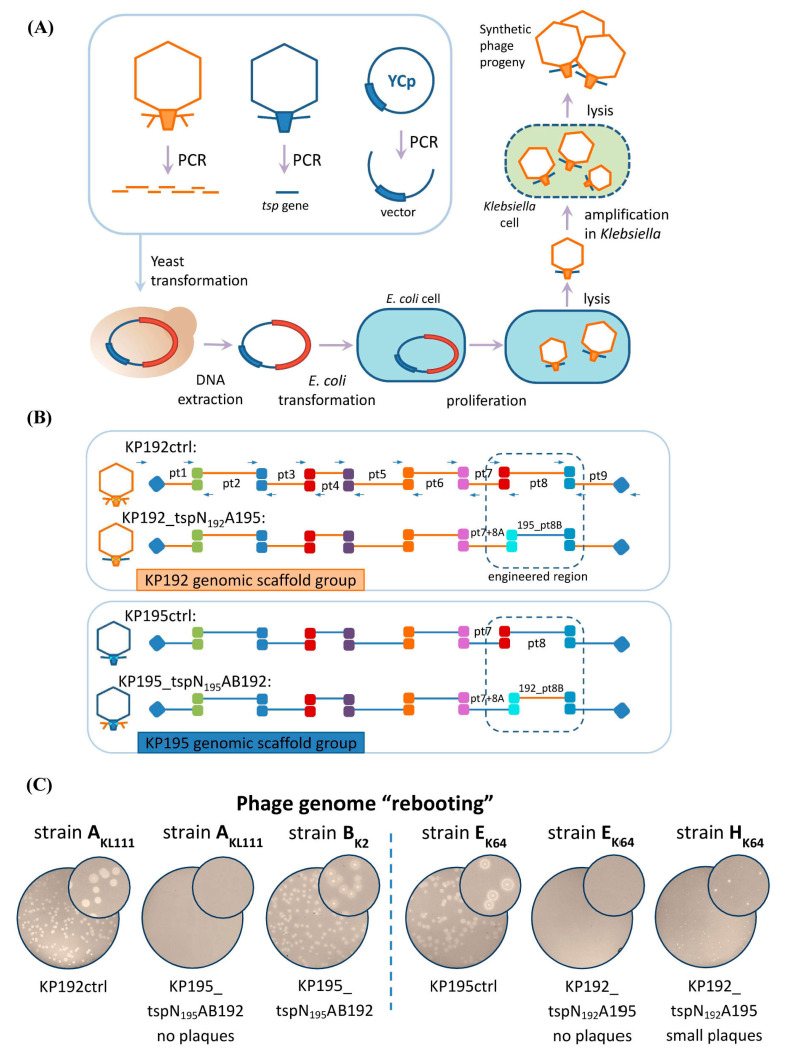
Assembly and “rebooting” of the synthetic phage genomes. (**A**) An outline of phage genome assembly in yeast and “rebooting” of phage genomes. “YCp”—yeast centromeric plasmid; “*tsp* gene”—a DNA fragment containing the gene(s) encoding tailspike protein(s). The red fragment represents an assembled phage genome integrated into a yeast plasmid. (**B**) Detailed diagram of the assembly of synthetic phage genomes (see also Appendix A). Primers are indicated by arrows. Regions of overlapping DNA fragments are shown as colored squares. Regions of overlap with the yeast centromeric plasmid are marked with blue diamonds. Parts 1 to 9 of the genome are designated “pt1”–“pt9”. The dashed lines indicate regions that differed between phages within the same scaffold group. (**C**) “Rebooting” of synthetic phage genomes. Plates containing a *Klebsiella* lawn and phage plaques are shown.

**Figure 4 ijms-26-11297-f004:**
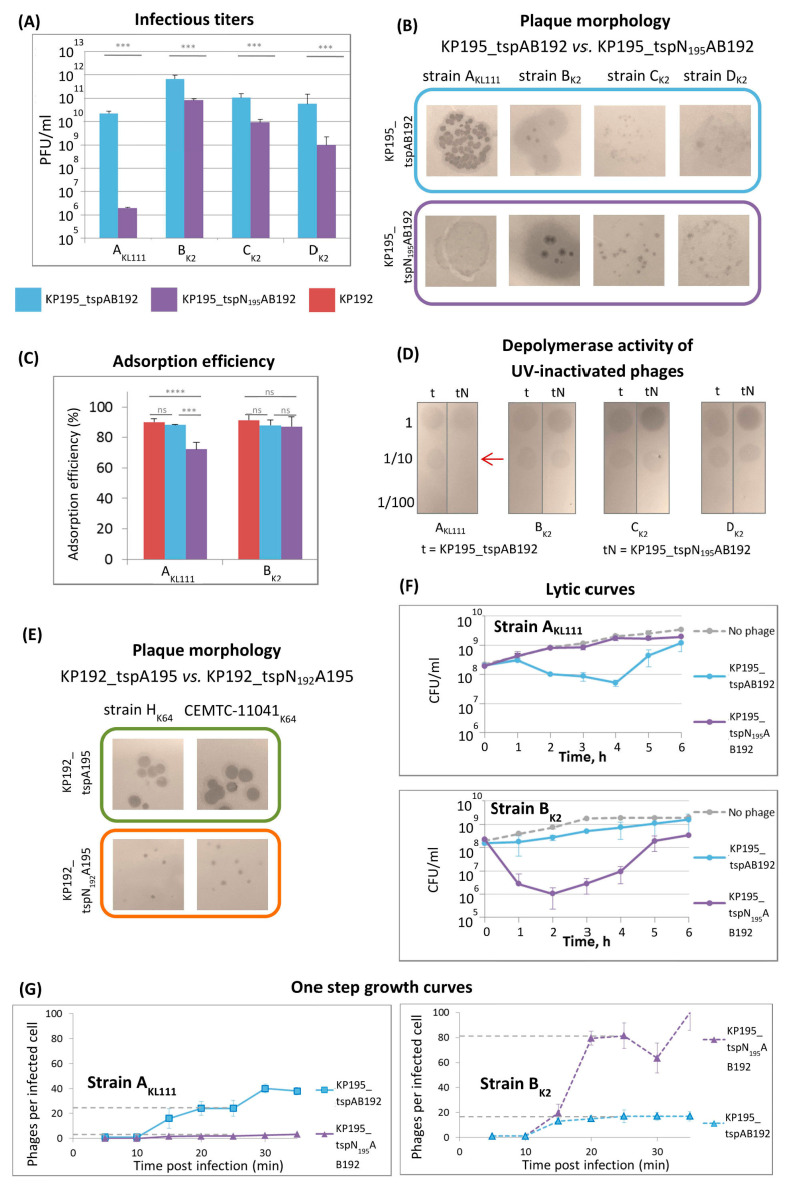
Differences in the genomic scaffolds of synthetic phages affect their replication efficiency. (**A**) Infectious titers of KP195_tspAB192 and KP195_tspN_195_AB192 phage samples. The phage particle concentration in the samples was the same (4.6 × 10^11^ PCLU/mL). Data from n = 3 independent experiments are represented as the mean ± SD. Statistical significance of log-transformed infectious titer values was determined using a one-way ANOVA with Tukey’s multiple comparison test (*** *p* < 0.001). (**B**) Plaque morphology. The most representative spots, clearly demonstrating the morphology of plaques, were selected. The size of the area shown is 12 × 12 mm. See also Appendix A for images of the whole plate surfaces and experiments using additional strains. (**C**) Adsorption efficiency. The fraction (%) of adsorbed phages following a 7 min adsorption. Test strains are indicated. Indicator strains are listed in Section 4. Data from n = 4 independent experiments are represented as the mean ± SD. Statistical significance was determined using a one-way ANOVA with Tukey’s multiple comparisons test (*** *p* < 0.001, **** *p* < 0.0001, “ns”—not significant). (**D**) Spot test of the depolymerase activity of UV-inactivated phage samples. Dilutions of the phage stocks are indicated on the left. The phage particle concentration in the stocks was the same (4.6 × 10^11^ PCLU/mL). (**E**) Plaque morphology for KP192_tspA195 and KP192_tspN_192_A195. The size of the area shown is 12 × 12 mm. See also Appendix A for experiments using additional strains. (**F**) Bacteria killing (lytic) curves; 1 PCLU of phage per 10 cells was used for infection. Data from n = 3 independent experiments are represented as the mean ± SD. (**G**) One-step growth curves. Data from n = 3 independent experiments are represented as the mean ± SD.

**Figure 5 ijms-26-11297-f005:**
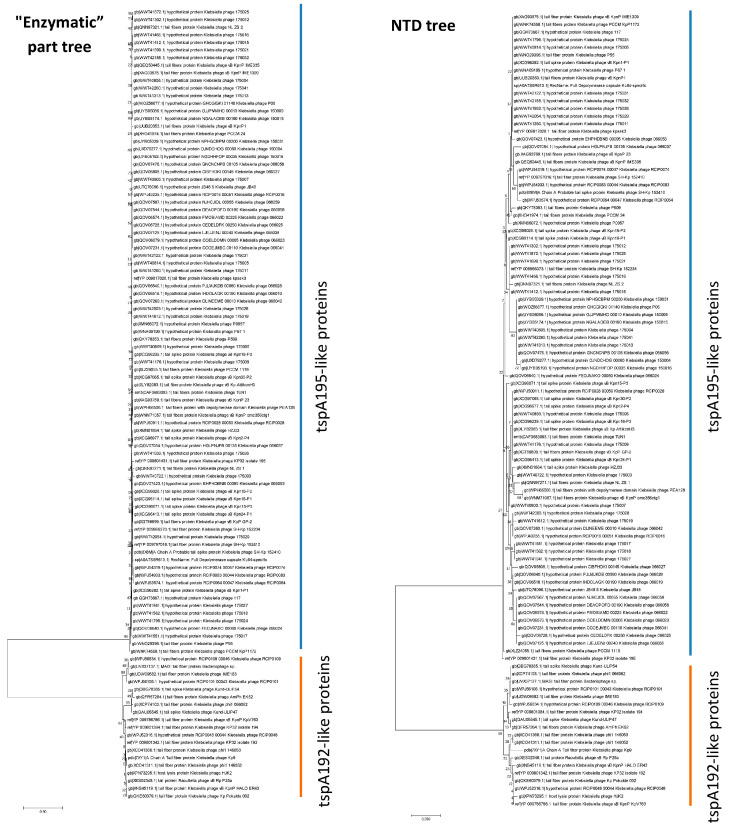
Phylogenetic trees constructed from the sequences of the enzymatic part (the entire tailspike protein except for the first 149 amino acid residues) or the N-terminal domain of tspA192-like and tspA195-like tailspikes. The numbers indicate the bootstrap values.

**Table 1 ijms-26-11297-t001:** Summary of the properties of the wild-type and synthetic phages.

	Phages with tspA192 and tspB192 Tailspikes	Phages with tspA195 Tailspikes
Name	KP192/KP192ctrl	KP195_tspAB192	KP195_tspN_195_AB192	KP195/KP195ctrl	KP192_tspA195	KP192_tspN_192_A195
Type	wild-type ^1^	synthetic	synthetic	wild-type ^1^	synthetic	synthetic
Pictogram	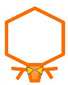	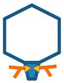	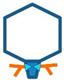	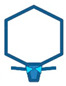	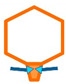	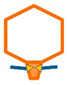
Capsular specificity	KL111 and K2	KL111 and K2	KL111 and K2	K64	K64	K64
Genomicscaffold	KP192	KP195	KP195	KP195	KP192	KP192
N-terminal domain of the type A tailspikes	N_192_	N_192_	N_195_	N_195_	N_195_	N_192_
NTD of the type A tailspikes is native to the gatekeeper and nozzle proteins	Yes	No	Yes	Yes	No	Yes
*Klebsiella* strain usedfor genome“rebooting”	A_KL111_	A_KL111_ ^2^	B_K2_	E_K64_	E_K64_ ^2^	H_K64_
Adsorption efficiency	A_KL111_ B_K2_	90 ± 2%91 ± 4%	A_KL111_ B_K2_	88 ± 1%88 ± 4%	A_KL111_ B_K2_	72 ± 5%87 ± 6%	—	—	—
Depolymerase activity of UV-inactivated phages	— ^3^	—	A_KL111_: weaker than that of KP195_tspAB192B_K2_:, C_K2,_ D_K2_:same as for KP195_tspAB192	—	—	—
Efficiency of planktonic culture lysis	A_KL111_: high(clear lysate) ^2^B_K2_: no lysis ^2^	A_KL111_: weak(turbid lysate)B_K2_: moderate (clear lysate at MOI = 0.1, turbid lysate at MOI = 0.01)	A_KL111_: no lysisB_K2_: high(clear lysate)	—	—	—
Burst size(phages/infected cell)	—	A_KL111_ B_K2_	~25~17	A_KL111_B_K2_	~3~80	—	—	—

^1^ Phages KP192 and KP195 were wild-type, while KP192ctrl and KP195ctrl were their synthetic analogs. ^2^ According to [14]. ^3^ “—“ not studied.

**Table 2 ijms-26-11297-t002:** Sequence conservation of the N-terminal domains of tspA192-like and tspA195-like tailspikes.

Tailspike Group	Number of Sequences	Capsular Specificity	Amino Acid Identity of NTD (Within Group)	Mean Distance (Within Group)	Amino Acid Identity of NTD (Between Groups)	Mean Distance (Between Groups)
tspA192-like proteins	19	KL111	≥95%	0.018	≤70%	0.403
tspA195-like proteins	85	K64	≥93%	0.026	≤70%	0.403

## Data Availability

All data generated in this study are available from the corresponding authors upon reasonable request.

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
