# Peer review of "The N-Terminal Domain of Tailspike Depolymerases Affects the Replication Efficiency of Synthetic Klebsiella Phages"

_ijms, 2025, doi:10.3390/ijms262311297_

Round 1
Reviewer 1 Report
Comments and Suggestions for Authors
A brief summary
The article is about engineering of synthetic bacteriophages of the genus Przondovirus that infect Klebsiella pneumoniae, with replaced genes encoding receptor-binding proteins (RBPs). The researchers suggest the N-terminal domain (NTD) to be not merely a structural adaptor but also a unit influencing binding and hydrolytic activity. In the article it is described that the chimeric tspA showed substantial reductions in depolymerase activity on specific capsular types. Herewith phylogenetic analysis showed a tight coevolution of the NTD and catalytic domains of tspA, manifested by the absence of natural hybrids and reflecting the importance of compatibility between these regions for phage fitness.
Thus the article raises interesting questions about the nature of tailspike functioning and the influence of the N-terminal domain on these processes. However, some questions still arise.
General concept comments
Considering that the discussion is centered around the structure of receptor-binding proteins and their structures, I would like the authors to describe the phages themselves in a little more detail in the introduction, their size, and show in which parts of the genome the proteins being studied are encoded.
In the “rebooting” section 2.2. wild types and modified ones are rebooted on different strains due to incapability of KP195_tspN195AB192 and KP192_tspN192A195 to grow on wild type hosts. But why not to use strains BK2 and HK64 for all of them?
It would be interesting to confirm the assumption about the influence of the N-terminus on the efficiency of the enzymatic part by obtaining recombinant proteins and comparing their activity.
More experiments are likely needed to provide more definitive statements and confirm or refute authors assumptions. For example, consider NTDs from other phages. Doubts are also heightened by the fact that the experiments were conducted on only a small sample of bacterial strains, 1-3 for each capsule type. Perhaps the characteristics of AKL111 are specific to this strain, and the effect on other strains of the same type will be different. I see this as the main drawback of the article and recommend that the authors test their phages on a larger sample of Klebsiella pneumoniae strains.
The links you provided (13) contain confirmation of successful replacement options. Could there be some difference in your approaches?
What parameters were used in paragraph 2.4 to determine similar sequences? What does this mean?
To improve understanding, I could recommend modifying Table 1 and supplementing it with detailed results of what is working and where. This will allow the reader to immediately assess the full picture of what is happening.
Also I can recommend authors the ddPCR method for measuring the concentration of phage particles, which can be used even at low concentrations.
Specific comments
On the fig 2 (C) it is not understandable where is a plate with a which capsule type. Every plate should be named for better understanding.
In all figures and tables, the phages should be named in the same way as in the text; it requires mental effort to constantly switch from KP195_tspN195AB192 to tspN195AB19.
In Figure 3(B), it's unclear what principle was used to select the dilution for visualization. The different units of measurement in parts (a) and (b) are also confusing; it would be great to standardize them to make things easier for the reader.
Reviewer 2 Report
Comments and Suggestions for Authors
The authors attempted to investigate the function of the N-terminal domain of tailspike in two Przondovirus phages, KP192 and KP195. These two phages were engineered to assemble chimeric tailspikes where their N-terminal domains were swapped. Further biological assays were performed to probe the change of replication and infection upon the tailspike engineering. The results indicate that the N-terminal domain of tailspike is essential for host recognition, consistent to its conservation within the same phylogenic group but not across different groups. Overall, I think this is a scientifically solid work that could be published. I only have a few minor questions/suggestions:
The x-axes of Figure 3F miss annotations.
Could the authors do the same one-step growth analysis (Figure 3G) for KL111 as well? Would it show the opposite result compared to Figure 3G?
The statement of lines 257-260 is not concise to me. What does it mean by ‘mismatch’? Does it suggest a change of binding between the chimeric tailspike and its neighboring proteins? If so, did the authors perform any biochemical or biophysical experiments to test their affinity? If not, it is not ‘apparent’ that a ‘mismatch’ or any conformational change appears in the engineered tail.
Round 2
Reviewer 1 Report
Comments and Suggestions for Authors
Dear authors, thank you for your responses and clarifications. This version of the article has become more transparent and understandable.